# Genomic Characterization and Genetic Profiles of *Salmonella* Gallinarum Strains Isolated from Layers with Fowl Typhoid in Colombia

**DOI:** 10.3390/genes14040823

**Published:** 2023-03-29

**Authors:** Ruy D. Chacón, Manuel Ramírez, Carmen L. Rodríguez-Cueva, Christian Sánchez, Wilma Ursula Quispe-Rojas, Claudete S. Astolfi-Ferreira, Antonio J. Piantino Ferreira

**Affiliations:** 1Department of Pathology, School of Veterinary Medicine, University of São Paulo, São Paulo 05508-270, Brazil; ruychaconv@usp.br (R.D.C.); csastolfi@gmail.com (C.S.A.-F.); 2Inter-Units Program in Biotechnology, University of São Paulo, São Paulo 05508-900, Brazil; 3Unidad de Bioinformática, Centro de Investigaciones Tecnológicas, Biomédicas y Medioambientales, Bellavista 07006, Peru; mramirez@citbm.pe; 4Laboratory of Biology and Molecular Genetics, Faculty of Veterinary Medicine, Universidad Nacional Mayor de San Marcos, Lima 15021, Peru; carmen.rodriguez12@unmsm.edu.pe; 5Department of Genetics, Physiology and Microbiology, Faculty of Biological Sciences, Complutense University of Madrid (UCM), 28040 Madrid, Spain; chrsan01@ucm.es; 6Laboratory of Molecular Microbiology and Biotechnology, Faculty of Biological Sciences, Universidad Nacional Mayor de San Marcos, Lima 15081, Peru; wilma.quispe@unmsm.edu.pe

**Keywords:** *Salmonella* Gallinarum, vaccine 9R, whole-genome sequencing, multilocus sequence typing, *Salmonella* pathogenicity island, prophage, virulence factors, antimicrobial resistance genes, plasmids, mobile genetic elements

## Abstract

*Salmonella* Gallinarum (SG) is the causative agent of fowl typhoid (FT), a disease that is harmful to the poultry industry. Despite sanitation and prophylactic measures, this pathogen is associated with frequent disease outbreaks in developing countries, causing high morbidity and mortality. We characterized the complete genome sequence of Colombian SG strains and then performed a comparative genome analysis with other SG strains found in different regions worldwide. Eight field strains of SG plus a 9R-derived vaccine were subjected to whole-genome sequencing (WGS) and bioinformatics analysis, and the results were used for subsequent molecular typing; virulome, resistome, and mobilome characterization; and a comparative genome study. We identified 26 chromosome-located resistance genes that mostly encode efflux pumps, and point mutations were found in gyrase genes (*gyrA* and *gyrB*), with the *gyrB* mutation S464T frequently found in the Colombian strains. Moreover, we detected 135 virulence genes, mainly in 15 different *Salmonella* pathogenicity islands (SPIs). We generated an SPI profile for SG, including C63PI, CS54, ssaD, SPI-1, SPI-2, SPI-3, SPI-4, SPI-5, SPI-6, SPI-9, SPI-10, SPI-11, SPI-12, SPI-13, and SPI-14. Regarding mobile genetic elements, we found the plasmids Col(pHAD28) and IncFII(S) in most of the strains and 13 different prophage sequences, indicating a frequently obtained profile that included the complete phage Gifsy_2 and incomplete phage sequences resembling Escher_500465_2, Shigel_SfIV, Entero_mEp237, and Salmon_SJ46. This study presents, for the first time, the genomic content of Colombian SG strains and a profile of the genetic elements frequently found in SG, which can be further studied to clarify the pathogenicity and evolutionary characteristics of this serotype.

## 1. Introduction

The genus *Salmonella* belongs to the family Enterobacteriaceae and includes more than 2600 different serotypes (or serovars) [1]. These Gram-negative bacilli are facultatively anaerobic and are critical for dangerous disease outbreaks in humans and animals [2]. In domestic birds, the serovar Gallinarum is host-specific and nonmotile and can be subclassified into the biovars Gallinarum (bvG, also called *S.* Gallinarum) and Pullorum (bvP, also called *S.* Pullorum) which are the causative agents of fowl typhoid (FT), which mainly affects mature birds and pullorum disease, which mainly affects young birds, respectively [3]. FT caused by *S.* Gallinarum (SG) produces an acute or chronic septicemic disease and is responsible for profound economic losses in certain countries [4]. It can be transmitted horizontally via contact with infected birds and contaminated farm materials and operators and vertically through fertilized eggs. Morbidity and mortality are usually high, affecting more than 90% of a flock [3,4]. A particular pathogenic characteristic of SG is the ability to infect and multiply within the mononuclear phagocyte system (MPS) of birds, eliciting a systemic infection that facilitates invasion and survival within macrophages [3,5].

*Salmonella* is endowed with machinery comprising genetic virulence factors. These factors are generally encoded in clusters and are transferred together via pathogenicity islands (SPIs), plasmids, and prophages. In *Salmonella*, twenty-four SPIs have been described to date, and their number and composition vary among serovars [6,7,8]. SPI-1 and SPI-2 are the two most studied SPIs and they play important roles in bacterial invasion and intracellular survival by activating secretion systems (e.g., the type 3 secretion system, or just T3SS) [3,5,9]. Plasmids are mobile genetic elements that increase the bacterial arsenal by carrying genes associated with antimicrobial resistance and virulence [8,10]. Additionally, integrated bacteriophages (prophages) carry genes involved in virulence and the T3SS [6,11,12]. The study of SPIs and virulence factors has shown differences in the gene content among pathogenic and non-pathogenic *Salmonella* [13]. The distribution of the virulence factors among different SG isolates could provide new insights into the mechanisms of SG pathogenesis.

Measures to control the incidence of FT are directed to prevention and include establishing efficient biosecurity procedures and enhancing intestinal immunity [3,14]. However, prophylactic vaccination is complementary and considerably effective. Live vaccines are more effective than inactivated vaccines, with the 9R strain established as the best protective and the most extensively used live oral or injected vaccine [4,15,16,17,18]. Genomic studies comparing 9R and field SG isolates could provide new insights into SG virulence.

Whole-genome sequencing (WGS) is increasingly being used to simultaneously identify the characteristics of pathogenic *Salmonella* via in silico molecular typing, the detection of antimicrobial resistance genes and virulence factors, and the identification of mobile genetic elements [2,19,20]. Thus, some genomic studies in SG have been published with a focus on comparisons against nontyphoid serovars [2,21], between Gallinarum and Pullorum biovars [22], or comparisons in strains belonging to serovar Gallinarum [23]. However, studies of SG field strains have been limited to a few countries and have never included Colombian SG strains [24,25,26].

Considering all the aforementioned factors, the complete genome sequence of eight field strains of SG isolated from birds infected during FT outbreaks in Colombia was characterized in this study. We identified genetic features related to the resistome, virulome, and mobilome of investigated SG genomes.

## 2. Materials and Methods

### 2.1. Bacterial Isolation

Fowl typhoid outbreaks occurred in layer breeders and layer hens in poultry farms in Bucaramanga, Colombia, in 2017. Details on four of these outbreaks were reported [27]. The mean weekly mortality rate was 8%. At necropsy, infected birds presented with a swollen and friable liver with necrotic foci, hepatomegaly, and splenomegaly. A pool comprising specific organs that had been collected from 5 different hens from the same flock was processed under sterile conditions. The homogenates were cultured in tetrathionate broth (Difco^TM^, BD, Le Pont de Claix, France) for 48 h at 37 °C and then cultured in MacConkey (BBL^TM^, BD, Le Pont de Claix, France) and xylose–lysine–Tergitol 4 (XLT4) agar (Difco^TM^, BD, Le Pont de Claix, France) for 24 h at 37 °C. Typical *Salmonella* colonies were subjected to biochemical testing using an API 20E kit (Biomerieux, Marcy l’Etoile, France). The farms, types, sources, and ages of the studied birds are indicated in Table 1.

### 2.2. DNA Extraction and PCR Identification

After biochemical identification, pure colonies were cultured in LB broth (Invitrogen, Carlsbad, CA, USA) for 18 h at 37 °C for DNA extraction using a ChargeSwitch genomic DNA (gDNA) mini bacteria kit (Thermo Fisher Scientific, Carlsbad, CA, USA). The DNA concentration was diluted to 0.5 ng/L based on the Qubit fluorometric quantitation and with a double-stranded DNA (dsDNA) broad-range (BR) assay kit (Life Technologies Corporation, Eugene, OR, USA). A multiplex PCR assay was performed to differentiate *Salmonella* biovars Gallinarum (SG), Pullorum (SP), and vaccine strain 9R [28].

### 2.3. Whole-Genome Sequencing, Assembly, and Annotation

Whole-genome sequencing of one isolate per farm plus an available commercial 9R-based SG vaccine (9R-Col) was performed. Libraries were prepared using a Nextera XT DNA library preparation kit (Illumina, Inc., San Diego, CA, USA), and sequencing was performed using a MiSeq sequencing system (Illumina, Inc.), with 500 cycles of amplification followed by paired-end sequencing (2 × 250-bp reads).

Before library assembly, the quality of raw reads was evaluated using FastQC v0.11.9 [29], and low-quality reads were excluded by Trimmomatic v0.36 [30]. The genomes were assembled de novo by using SPAdes v3.15 [31], excluding contigs smaller than 500 bp and coverage means smaller than 2. QUAST v4.1 [32] was used to determine the quality of the assembly stats (genome size, number of contigs, N50 value, and G+C content). Finally, we annotated the assembled genomes using the NCBI Prokaryotic Genome Annotation Pipeline (PGAP v. 6.3) [33]. The raw sequenced reads were deposited into the National Center for Biotechnology Information Sequence Read Archive database (NCBI SRA). The assembled genome sequences were deposited in GenBank under BioProject number PRJNA483415. We also retrieved 11 additional SG genomes with related available information on the country and year of isolation from the SRA and GenBank databases. Three of these retrieved genomes (SA68, 287/91, and SG9-1955) correspond to complete genomes. The information on the SG strains used for this study is indicated in Table 2.

### 2.4. Bioinformatics Analysis: Phylogenetics. Molecular Typing. Virulome, Resistome, and Mobile Genetic Element Characterization; and Comparative Genome Analyses

Comparative genomics analyses were performed with the 20 SG strains mentioned above. Initially, we used Snippy v4.6.0 (https://github.com/tseemann/snippy, accessed on 1 November 2022) to produce an alignment of “core SNPs (single nucleotide polymorphisms)” by setting the strain 287/91 (GenBank: NC_011274) as the reference. We used this core SNP genome alignment to build a phylogenic tree. The best-fit substitution model for the phylogenetic analysis was estimated with ModelTest-NG v0.1.7 [34] based on Akaike information criteria (AIC). To infer phylogenetic relationships among the sequences, a maximum likelihood tree was constructed with RaxML-NG v1.1.0 [35] using the TVM nucleotide substitution model, and nodal supports were estimated with 1000 bootstrap replicates. The tree was visualized and annotated using ITOL v6 [36].

In silico serotyping was performed with SISTR v1.0.1. [37]. We used available pipelines (https://github.com/tseemann, accessed on 1 November 2022) for multilocus sequence typing (MLST) using mlst 2.11 [38]. The presence of acquired antibiotic resistance genes and virulence factors was determined with ABRicate v1.0.1 according to the Comprehensive Antibiotic Resistance Database (CARD) [39] and the Virulence Factor Database (VFDB) [40]. PlasmidFinder 2.1 was used to detect plasmid sequences [41].

We also used the available online tools in the Center for Genomic Epidemiology (CGE) platform (https://www.genomicepidemiology.org/, accessed on 1 November 2022), namely, by SPIFinder v2.0, to identify *Salmonella* pathogenicity islands [42] and to identify chromosomal point mutations associated with acquired antimicrobial resistance by ResFinder v4.1 [43]. For SPI identification, we chose a threshold of 95% for a minimum identity and three different thresholds for the minimum coverage percentage of an SPI. We defined the results based on either complete coverage (=100%) or incomplete coverage (100% > coverage > 60%, and 60% ≥ coverage > 20%). We performed additional manual BLASTN searches to reconfirm the presence of SPIs.

The detection of prophages was performed with the Phaster search tool [44]. Genomes were mapped against the reference genome strain 287/91 and visualized using BRIG v0.95 [45].

## 3. Results

### 3.1. Genome Assembly, Annotation, and Molecular Typing

Colombian SG strains were sequenced, and the resultant assemblies were comprised between ~4.70 and 4.77 Mb with 34 to 175 contigs, the genome depth ranged from 58× to 77×, and the GC content ranged from 50.1 to 53.1% (Table 3). Genome annotation was performed, and between 4505 and 4593 CDSs were thus identified. These results were in accordance with those obtained from an analysis of the genomic data obtained for other SG strains. In silico serotyping using SISTR was performed to classify all the strains in the D1 serogroup and serovar Gallinarum, with O antigens 1, 9, and 12. The molecular typing of these sequenced and analyzed strains revealed a high frequency of ST 78 among the samples, except for the reference strain 287/91, which was typed as ST 331 (Table 3).

### 3.2. Antimicrobial Resistance Genes

We used an alignment of the core SNPs involving 1570 base pairs to cluster the SG strains. Because the raw data of three retrieved genomes (SA68, 287/91, and SG9-1955) were not available for assembly, we selected core SNP alignment to build the phylogeny. The phylogenetic tree inferred from this alignment showed clustering partially related to geographic origin. Specifically, all eight Colombian field strains were grouped into one cluster (Figure 1). The 9R-Col vaccine was clustered, as expected, with the other SG9 strains and with two field strains from France and Brazil. The human-origin strains were clustered together, and the African and Indian strains were isolated in separate basal branches.

We used the Comprehensive Antibiotic Resistance Database (CARD) [39] to search for the genes associated with antimicrobial resistance (AMR). We chose the CARD because it included both chromosomal and acquired genes. On the other hand, we used the ResFinder v4.1 tool [43] exclusively to look for chromosomal point mutations associated with AMR. The explored AMR genes included *parC*, *parE*, *gyrA*, *gyrB*, *pmrA*, *pmrB*, *acrB*, and *16S_rrsD*.

Regarding the presence of genes, we consistently found the same 26 chromosomal genes in almost all SG strains, except strain 4295/02, which lacked the *marA* gene, a global activator protein that, in addition to inducing the MDR efflux pump *AcrAB*, downregulates the synthesis of the porin *OmpF* (Figure 1). Most of these genes encode efflux pump complexes and belong to the following AMR gene families: the major facilitator superfamily—MFS (*emrAB*, *emR*, *tolC*, and *hns*), the resistance–nodulation–division superfamily—RND (*acrAB*, *sdiA*, *marA*, *acrD*, *baeR*, *cpxR*, *mdtBC*, *mdsAB*, *golS*, *cpxA*, and *crp*), the multidrug and toxic compound extrusion family—MATE (*mdtK*), and the ATP-binding cassette family—ABC (*mbsA* and *YojI*). Additionally, the two-component regulatory system *kdpDE*, general bacterial porin *ompA*, undecaprenyl pyrophosphate-related protein *bacA*, and *aac*(*6′*)-*Iaa*, which belongs to the AAC(6′) aminoglycoside acetyltransferase family, were identified. These genes have been previously associated with resistance to antiseptics, disinfecting compounds, and a broad variety of antimicrobial classes, including cephems, monobactams, penems, penicillins, aminoglycosides, ansamycins, fluoroquinolones, macrolides, nitroimidazoles, peptides, phenicols, tetracyclines, aminocoumarin, anthracyclines, protonophores, and thiolactones (Appendix A) [39].

Moreover, the ResFinder tool revealed genes with aminoglycoside resistance. The *aac*(*6′*)-*Iaa* gene, which potentially confers resistance to amikacin and tobramycin, was found in all the strains. In addition, the strain CDC 4801/72 carried the plasmid-encoded genes *aph*(*3”*)-*Ib* and *aph*(*6*)-*Id*, which confers resistance to streptomycin (Figure 2).

Regarding point mutations, we found resistance to quinolones in gyrase genes. The mutation S464T in *gyrB*, which confers resistance to ciprofloxacin, was found only in six Colombian field strains. In the *gyrA* gene, we found four variant mutations that confer resistance to nalidixic acid and ciprofloxacin. These mutations were S83F (in the Brazilian strain 287/91), D87G (in the French strain 11CEB2315SAL), D87N (in the Tanzanian strain ST 78), and D87Y (in the Indian strain VTCCBAA614) (Figure 2).

### 3.3. Virulence Genes

We identified fifteen different SPIs among the analyzed SG strains through SPIFinder v2.0 (Figure 2, Appendix A). The number of SPIs varied from 13 to 15 (15 in all Colombian strains). Regarding the completeness of SPIs, we identified the following seven complete SPIs (in ≥90% of the studied strains): SPI-1, SPI-2, SPI-3, SPI-13, SPI-14, C63PI, and the putative SPI protein ssaD (Figure 3). In addition, we identified eight incomplete SPIs in the studied strains: SPI-4, SPI-5, SPI-6, SPI-9, SPI-10, SPI-11, SPI-12, and CS54. Surprisingly, the strain with the fewest SPIs (=13), the Brazilian strain 4295/02 isolated in 2002, lacked SPI-1, one of the most important pathogenicity islands in *Salmonella*, as well as C63PI.

A detailed search for specific virulence factors led to the identification of between 130 and 135 genes in most of the strains, except for the Brazilian strain 4295/02, which presented only 94 genes (this was mainly due to the absence of SPI-1 genes) (Figure 4, Appendix A). All the SG strains carried the tssJLM gene, which belongs to the type VI secretion system (T6SS). However, the strain CDC 4801/72 isolated from humans in 1972 carried three additional genes (IDs: STM0276, STM0284, and STM0287) related to the T6SS.

When comparing the Colombian strains with 9R-Col (a 9R-based vaccine), we identified some differences in the gene content. The 9R-Col lacked three genes that were present in all the field strains: *invJ* (an SPI-1-encoded T3SS needle-length regulator), *spvD* (a plasmid-encoded T3SS effector cysteine hydrolase), and *ssaK* (an SPI-2-encoded T3SS stator). Moreover, 9R-Col carried the gene *shdA* (an adhesin and AIDA autotransporter-like protein), which was absent in all the Colombian field strains but was found in other strains. Finally, the strains Buga, FTA303, GDX-03965-17, and GDX-03999-17 lacked the gene *ratB* (an adhesin and CS54-encoded putative outer membrane protein), while the strains FTA303, GDX-2, GDX-7, and GDX-03999-17 lacked the gene *spvB* (a plasmid-encoded toxin and T3SS effector with ADP-ribosylation activity). In addition, the strain GDX-03999-17 lacked the gene *sseL* (a T3SS effector with deubiquitinase activity) (Figure 4).

We detected four plasmid-encoded virulence factors: (1) the gene *mig*-*5* (a putative carbonic anhydrase with antimicrobial activity) was found in several strains, including all the Colombian SG strains, (2) the gene *spvB* encoding a toxin was found in some field strains, (3) the gene *spvC*, a T3SS effector encoding a phosphothreonine lyase, and (4) the gene *spvD*, a T3SS effector encoding a cysteine hydrolase (Figure 4).

Other genes exclusively absent in some strains were *avrA*, *sptP*, and *sipABCD*, and all the SPI-1-encoded genes were absent in the Brazilian strain 4295/02. The genes *sopD2* and *sseK1* were absent in the Indian strain VTCCBAA614. The *sseL* gene was absent in the strains GDX-03399-17 and 4295/02. The *sspH2* gene was absent in the strains ST572 and 4295/02 (Figure 4).

### 3.4. Mobile Genetic Elements

The search for plasmids revealed a high prevalence of Col(pHAD28) (68,42%; 13/19 strains) and IncFII(S) (73,68%; 14/19 strains) in SG strains (Figure 2). On the other hand, the strain CDC 4801/72 isolated from humans in 1972 presented only the plasmid IncI1-I(α). In four strains (4295/02, ST572, 11CEB2315SAL, and SARB21), no plasmid replicon was detected.

We explored the prophage diversity and profiles using the PHAge Search Tool Enhanced Release (PHASTER). According to the related database, thirteen most common phage (MCP) references (phages with the highest number of proteins most similar to those found in the query genome) were identified (Figure 2). The number of prophages by strain varied from three to six (Appendix A). A completeness evaluation showed a maximum of two intact prophages in some strains, while the other strains carried only one prophage, Gifsy_2. The other intact prophages detected were Salmon_vB_SenS_Ent2 (in strain FTA303), Salmon_Fels_1 (in strain 287/91), Salmon_Fels_2 (in strain ST 78 Tanzania), and Salmon_SEN8 (in strain ST572). Regarding the abundance of each MCP, intact prophages Gifsy_2 and incomplete prophages Escher_500465_2 were found in 100% (20/20) of the strains. Other frequently detected MCPs were incomplete Shigel_SfIV in 90% (18/20), incomplete Entero_mEp237 in 75% (15/20), and incomplete Salmon_SJ46 in 60% (12/20) of the strains. The remainder of the MCPs detected were found in only one strain each.

To gain a deeper insight, we evaluated the genetic content of the most frequent MCPs and then searched for the genes associated with virulence factors or antimicrobial resistance in the SG strain GDX58 as it presented the most prophages among the Colombian strains (Figure 5). The complete prophage Gifsy_2 carried the virulence factors *sopE* (encoding an SPI-1 type III secretion system guanine nucleotide exchange factor), *sodC1* (encoding superoxide dismutase [Cu-Zn]), *ail* (encoding the attachment invasion locus protein), *clpP* (encoding an ATP-dependent Clp protease proteolytic subunit), and *yjcS* (encoding the putative alkyl/arylsulfatase YjcS), a gene associated with resistance to sodium dodecyl sulfate (SDS). In addition, incomplete MCP Salmon_SJ46 presented the virulence factor *virB* (encoding the virulence regulon transcriptional activator VirB) and the genes *umuC* (encoding protein UmuC) and *umuD* (encoding protein UmuD) associated with UV protection. MCP Shigel_SfIV presented the virulence factors *yfdH* (encoding a prophage bactoprenol glucosyl transferase) and *yfdG* (encoding a prophage bactoprenol-linked glucose translocase/flippase), as well as genes potentially conferring resistance to copper *pcoE* (encoding the putative copper-binding protein PcoE) and reactive chlorine species (RCS) stress resistance genes *cusS* (encoding the sensor histidine kinase CusS) and *rclR* (encoding the RCS-specific HTH-type transcriptional activator RclR). MCP Entero_mEp237 presented the gene *yjcS* (encoding the putative alkyl/arylsulfatase YjcS) with associated resistance to SDS. MCP Escher_500465_2 presented the gene *yfcJ* (encoding the putative MFS-type transporter YfcJ).

## 4. Discussion

According to the U.S. Department of Agriculture, Latin American countries, especially Brazil, Mexico, Argentina, Colombia, and Peru, contribute significantly to global chicken meat production. In fact, Colombia was ranked among the top ten world producers in 2022, with 1.88 million tons [46]. Despite evidence showing that intermittent outbreaks of FT in Colombia reported in recent years caused significant economic impacts, in-depth genetic and genomic characterization studies of SG have been scarce [27,47]. Thus, our aim for this study was to describe the genomic characteristics of eight SG Colombian field strains and compare them against a 9R-based vaccine and strains from other countries. We performed molecular typing and explored the resistome, virulome, mobilome, and additional features that may lead to phenotypic and pathological differences among strains.

Bacteria engage a variety of mechanisms to resist the action of antibiotics. For some bacteria, the balance in membrane permeability/impermeability plays a key role in limiting the uptake of antimicrobial compounds and the subsequent diffusion and efflux that reduces the intracellular concentration [48]. We found 26 chromosomal genes in all the SG strains, with most encoding efflux pumps, associated with resistance to 16 classes of antimicrobial agents as well as antiseptics and disinfectants. These results are similar to previous WGS-based studies of chromosome-encoded genes [49,50,51]. The bacterial chromosome-encoded systems involved in AMR typically confer a low level of resistance, but they are important as possible ‘stepping stones’ to higher levels of resistance [52]. Several resistance mechanisms may lead to high-level antimicrobial resistance via the action of multidrug efflux pumps, mutations in gyrases and topoisomerases, and decreased outer membrane permeability [53,54]. Our results are in concordance with those of a previous report [51] and show an SG profile comprising conserved chromosomal genes associated with AMR. The impact of mutations and polymorphisms on the expression of these genes needs to be explored.

Gyrases are type II topoisomerases that modulate dsDNA topology and maintain chromosomes in a loosely wound state. They form heterotetramers composed of two *gyrA* and two *gyrB* chains. *GyrA* carries an active tyrosine site through which a transient covalent intermediate with DNA is formed, while *gyrB* binds cofactors and catalyzes ATP hydrolysis. In *Salmonella*, quinolone resistance has been attributed to point mutations in gyrase genes, mainly in a region termed the quinolone resistance-determining region (QRDR), which is located between amino acids 67 and 106 in *gyrA* [54]. We detected the *gyrA* mutations S83F, D87G, D87N, and D87Y in SG strains isolated from different continents. These mutations have been associated with quinolone resistance in *Salmonella* [54] and have been reported in SGs showing AMR to quinolones [55,56]. In addition, we detected the *gyrB* S464T mutation in several SG Colombian strains. This mutation has been reported only in a clinical *S.* Typhimurium strain, which shows resistance to ciprofloxacin [57].

Aminoglycoside resistance requires careful consideration. AMR genes against aminoglycosides are the most diverse and the most frequently identified in *Salmonella* field strains [58]. Moreover, the *aac*(*6′*)-*Iaa* gene has been frequently detected, and we confirmed that it was carried in all the analyzed SG strains. However, this gene has been reported to be a cryptic gene, with a mutation required to activate it [59]. On the other hand, horizontally acquired AMR genes were found in only one strain, CDC 4801/72, in the plasmid-encoded genes *aph*(*3”*)-*Ib* and *aph*(*6*)-*Id*, which confers resistance to streptomycin. This finding is interesting, as this strain was isolated from humans. The possibility of these genes being transferred to SGs should not be disregarded. In recent years, an increase in AMR against aminoglycosides and quinolones has been observed in SG [60]. The indiscriminate use of antimicrobial drugs in poultry farms, still being practiced in some countries, is considered to be responsible for this resistance and is a concern with respect to animal and human health [4,61].

SPIs are horizontally acquired loci encoding genes that facilitate several virulence mechanisms. While some of these SPIs are ubiquitous and distributed among all *Salmonella* species, others are associated with specific serotypes, revealing fitness advantages and host specificities [7]. We identified 15 different SPIs (C63PI, CS54, ssaD, SPI-1, SPI-2, SPI-3, SPI-4, SPI-5, SPI-7, SPI-9, SPI-10, SPI-11, SPI-12, SPI-13, and SPI-14) identified in most of the SG strains, which indicate that they are representative of the SG biovar profile. SPI-1 and SPI-2 are the most extensively studied islands, and they encode the T3SS and are fundamental to host cell invasion and intracellular survival, respectively [5,7,9]. Interestingly, the Brazilian strain 4295/02 lacked SPI-1, one of Salmonella’s most important pathogenicity islands. The genome sequence of this strain was retrieved from GenBank, and the isolation source was indicated as a sample of layer chicken litter via drag swabbing. An SPI-1 mutant SG showed decreased invasiveness of avian cells in vitro, but its ability to persist within chicken macrophages did not differ from that of the wild type [62].

It is not possible to rule out the circulation of a less virulent SG lineage potentially adapted to the poultry environment. Other pathogenicity islands consistently detected in SG were C63PI, which carries an iron uptake system, and the T3SS protein ssaD [63]. The CS54 island (also recognized as SPI-24), encodes intestinal colonization and persistence determinants [7]. As evolutionarily conserved elements, SPIs are susceptible to horizontal transfer and mutations [64]. Incomplete SPIs may be a consequence of constant evolution and host adaptation, including insertions and deletions that impair function or become pseudogenes. We have identified some incomplete SPIs in SG. Variations in the SPI content have been previously described, and SPIs may be differentially distributed among *Salmonella* serovars [65,66]. Our findings regarding virulence factors and SPIs are in concordance with previous studies [2,21,22,23]. The functional analysis of each SPI in SG should be performed to better understand the evolutionary processes and impacts of this serovar.

Protection against SG is globally covered with 9R vaccines. This strain is defective in O-side chain repeats in lipopolysaccharide (LPS) because of a mutation in the *rfaJ* gene, and the attenuated expression of this gene was initially attributed to this mutation [67]. In this study, we identified a 9R-Col vaccine lacking three virulence genes (*spvD*, *invJ*, and *ssaK*). The *spvD* gene has been previously reported to be deleted in a 9R strain [67]. Our results agreed with this finding, as all the 9R-related strains analyzed in this study (9R-Col, SG9-1955, and SG9-1995) lacked this gene (Figure 4). This plasmid-encoded gene is a T3SS effector of SPI-1 and SPI-2 and contributes to systemic infection in mice by inhibiting the NF-κB pathway and the secretion of proinflammatory cytokines [68]. The *invJ gene* is a T3SS encoded by SPI-1 and is essential for the projection of the needle length of the injectisome complex, which in turn is relevant to the protein secretion and bacterial invasion of epithelial cells [69,70]. The *ssaK* gene is a T3SS encoded by SPI-2 and is required to inject effector proteins into host cells to facilitate intramacrophage survival and virulence levels in an animal host [71]. Interestingly, a comparative proteomic study with wild-type and live 9R vaccines identified 42 virulence genes, including *invJ* and *ssaK*, that were downregulated in the vaccine strain [72]. Our results reinforce the notion that the attenuation of SG 9R strains may be a result of combined defects in virulence factors [72]. Additional studies exploring point mutations in 9R-based vaccines may further clarify the picture.

Plasmid replicons are mobile genetic elements capable of conferring or increasing bacterial virulence and resistance to antimicrobials. We identified two frequent plasmids distributed among SG strains. The most frequently distributed, the small plasmid Col(pHAD28) (GenBank: KU674895), was discovered in avian-origin *S. Hadar* and was associated with quinolone resistance because it carries the *qnrB19* gene [73]. However, the SG Col(pHAD28) variants found in this study lacked this gene. Interplasmid recombination is presumed to be the event underlying the formation of these variants [73]. The second most frequently distributed plasmid was identified as IncFII(S). According to the reference genome (strain 287/91), this plasmid has been named pSG (GenBank: HG970001), which is a virulence plasmid specific to the *S.* Gallinarum serotype. This plasmid has been characterized based on its *Salmonella* plasmid virulence locus (*spv*), the expression of which contributes to systemic virulence and intramacrophage survival [66].

Temperate phages (prophages) are viruses that infect bacteria and persist benignly within the chromosomal or plasmid DNA of a bacterial host. Complete or cryptic (defective) prophages can carry genes that contribute to the bacterial acquisition of antimicrobial resistance and the adaptation to new environments or hosts, increasing the virulence and pathogenicity of the bacteria [44]. We found 13 different MCPs distributed heterogeneously among the studied SG strains. No geographical or temporal relationship was observed concerning the composition or distribution of these MCPs. However, one complete prophage (Gifsy_2) and four cryptic prophages (Escher_500465_2, Shigel_SfIV, Entero_mEp237, and Salmon_SJ46) were frequently found in SG. The composition and diversity of the prophage sequence profiles and identified cargo genes may be indicative of specific serotypes, and therefore may contribute to subtyping approaches [74,75]. Interestingly, the abovementioned MCPs carried certain virulence factors as well as genes with the potential to confer resistance to chemical and physical stressors. The abundant diversity and insertion of phages into the SG genome may contribute to the pathogenicity and host adaptation of this chicken-associated serovar [66].

The genomic characteristics of eight SG Colombian field strains showed that they belong to a different genetic cluster than SG vaccine strains and field strains from other countries. Specific differences were also observed in the gyrase B gene which carried an infrequent point mutation (S464T) which is associated with resistance to ciprofloxacin [57]. Some virulence-associated genes were irregularly distributed in Colombian strains compared to other SG strains. We detected the plasmid-encoded virulence genes *spvC*, *spvD*, and *mig*-*5* in all Colombian strains, while they were less frequent in the other SG strains. *SpvCD* genes, which are T3SS effectors, were found to be expressed by *Salmonella* when ingested by macrophages, contributing to their survival and proliferation [10]. The *mig*-*5* gene has been associated with the resistance of *Salmonella* to the host complement system [10]. Conversely, the genes *ratB* and *shdA* were poorly detected or absent in the Colombian strains of GS, while they were more frequent in the other strains. These genes encode non-fimbrial adhesins located in the pathogenicity island CS54 and are involved in intestinal colonization and persistence in the host [76]. A particular case was the absence of the gene *sseL* in one Colombian strain. *SseL* is a T3SS effector with deubiquitinase activity required for the *Salmonella*-induced delayed cytotoxicity of the macrophages [77]. Salmonella gain and lose virulence factors over time as adaptive mechanisms to new hosts or environments [78,79]. On the other hand, the genomic profiles of SPIs, chromosomal genes associated with antimicrobial resistance, plasmids, and prophage sequences revealed relative commonalities in these important genetic features among Colombian strains and the SG isolated from other countries. These findings highlight the importance of these elements for the survival and pathogenicity of SG [6,7,8,10].

WGS is a powerful tool for detecting many diverse virulence factors. The use of bioinformatic tools such as ABRicate and SPIFinder [42] integrated into platforms and databases such as the Center for Genomic Epidemiology (CGE) and the Virulence Factor Database (VFDB) [40] in this study helped us to understand the mechanisms underlying SG virulence better. Defining the most important genetic traits that confer virulence and resistance to SG is essential. The knowledge of these features can allow us to predict the level of virulence and resistance, and thus develop strategies to reduce it.

## 5. Conclusions

The genomic characterization of Colombian SG isolates shows a high genetic similarity between them, but differences with other SG vaccine or field strains isolated in other countries. However, all the strains belonging to the SG serotype maintain a relatively conserved profile of pathogenicity islands, virulence factors, efflux pumps, and prophages. On the other hand, the comparison of the presence or absence of virulence factors between field strains and a 9R vaccine strain allows new perspectives to emerge on the pathogenicity of SG. Finally, the information obtained in this study may help to better understand the mechanisms underlying SG virulence and resistance.

## Figures and Tables

**Figure 1 genes-14-00823-f001:**
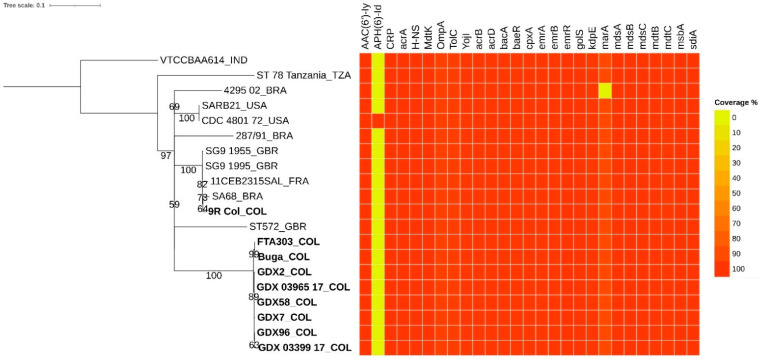
Phylogenetic tree based on a core SNP alignment was performed with RaxML-NG v1.1.0 [35] based on the TVM model and 1000 bootstrap replications. **Left**: The tree was visualized and annotated using ITOL v6 [36]. **Right**: Chromosomal genes associated with antimicrobial resistance (AMR) in the studied SG strains. The color range bar represents the % coverage concerning the CARD reference. The % coverage values range from 0% (yellow) to 100% (red).

**Figure 2 genes-14-00823-f002:**
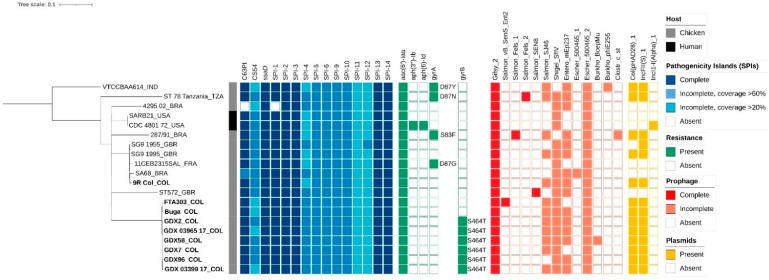
Genomic features of the studied *S. Gallinarum* strains. From left to right: phylogenetic tree, host, *Salmonella* pathogenicity islands, resistance genes, prophages, and plasmids.

**Figure 3 genes-14-00823-f003:**
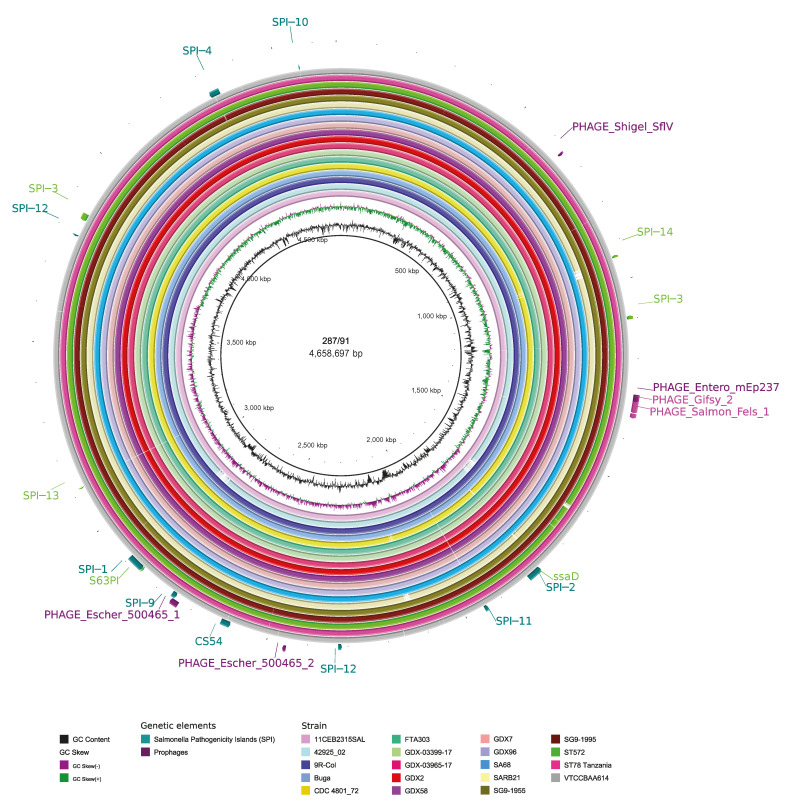
Genome comparison of the studied *S. Gallinarum* strains compared to the reference strain (287/91). GC skew and GC content, shown in the second and third rings, respectively, are representative of the reference genome. The other rings represent each analyzed genome. Annotated genetic features outside the rings are presented as a complete SPI (green), an incomplete SPI (lime), a complete prophage (purple), and an incomplete prophage (magenta). The data were visualized using BRIG v0.95 [45].

**Figure 4 genes-14-00823-f004:**
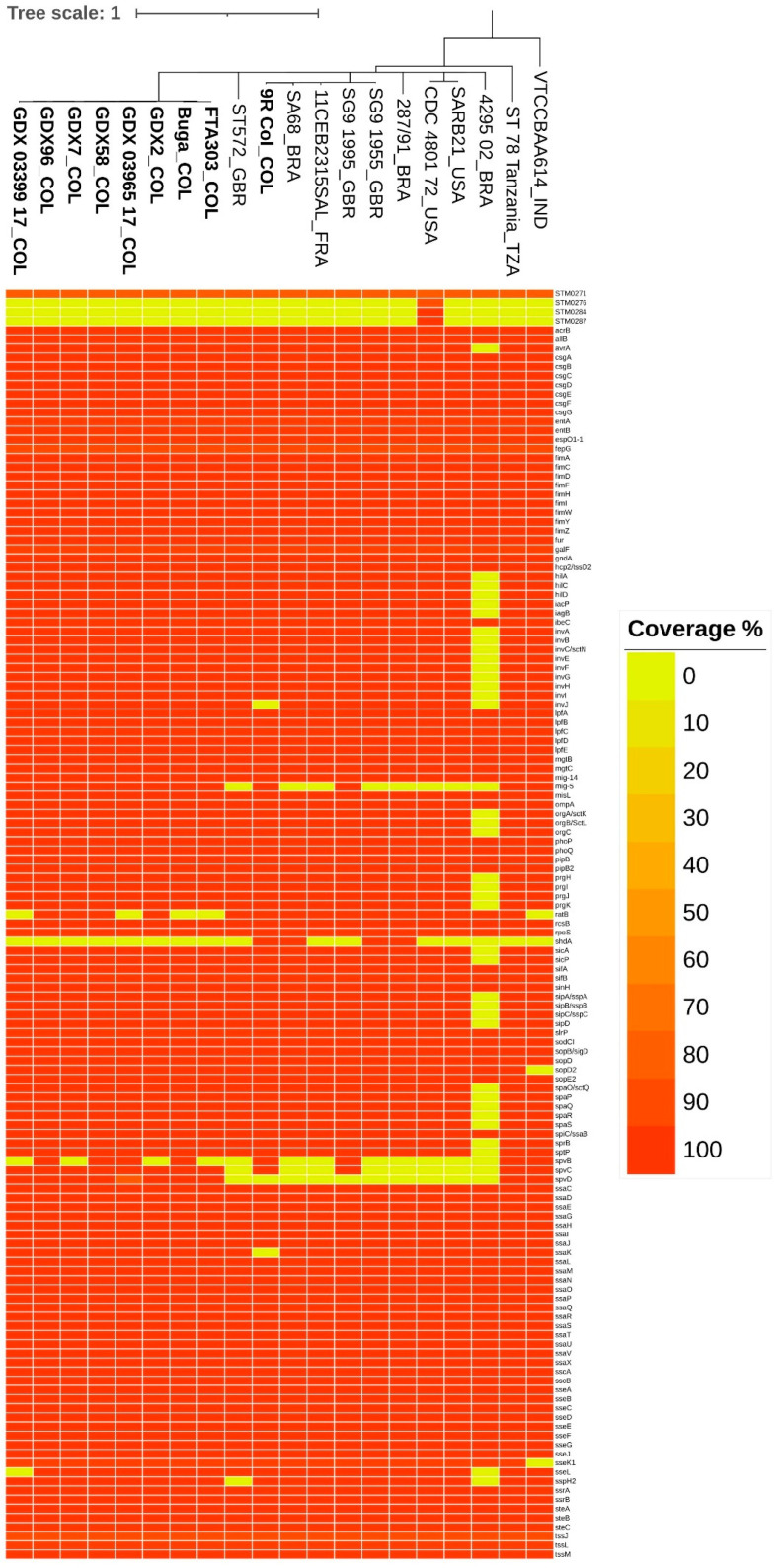
Genes defined as virulence factors identified in the studied SG strains. The color range bar represents the % coverage concerning the VFDB reference. The % coverage values range from 0% (yellow) to 100% (red).

**Figure 5 genes-14-00823-f005:**
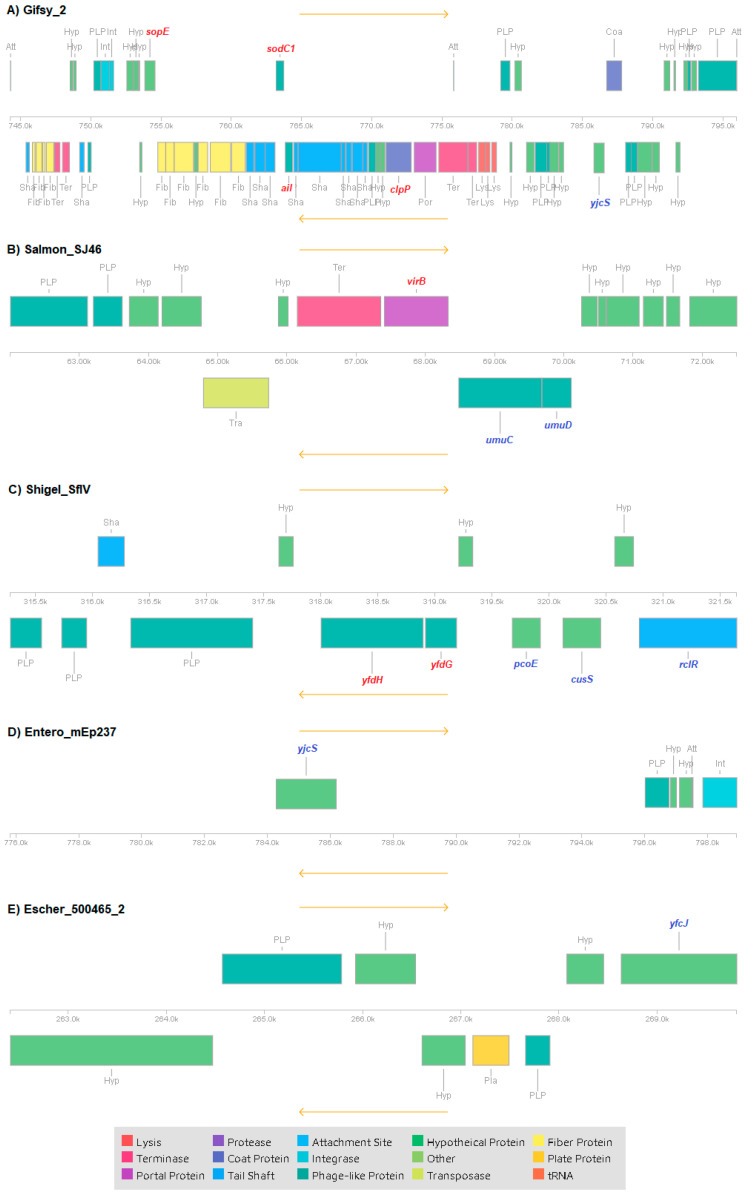
Genetic organization of the most common phages (MCPs) frequently detected in SG. (**A**) Gifsy_2, (**B**) Salmon_SJ46, (**C**) Shigel_SfIV, (**D**) Entero_mEp237, and (**E**) Escher_500465_2. Virulence and resistance genes are identified in red and blue, respectively. This figure was edited from a file generated using the Phaster tool [44].

**Table 1 genes-14-00823-t001:** Strains isolated in this study.

Farm	Isolated Strain	Source	Type of Bird	Age (Weeks)
1	Buga	Liver	Layer breeder	1
2	GDX7	Liver	Layer breeder	13
3	FTA303	Liver	Layer breeder	15
4	GDX58	Liver	Layer breeder	25
5	GDX2	Liver	Layer hen	1
6	GDX96	Liver	Layer hen	11
7	GDX-03965-17	Brain	Layer hen	23
8	GDX-03399-17	Coelomic cavity	Layer hen	63

**Table 2 genes-14-00823-t002:** Information on the strains used in this study.

Strain	Country	Year Isolated	SRA Accession No.
Buga	Colombia	2017	SRR8521219
GDX7	Colombia	2017	SRR8521149
FTA303	Colombia	2017	SRR8521220
GDX58	Colombia	2017	SRR8521221
GDX2	Colombia	2017	SRR22245387
GDX96	Colombia	2017	SRR22245388
GDX-03965-17	Colombia	2017	SRR22245385
GDX-03399-17	Colombia	2017	SRR22245386
9R-Col	Colombia	2017	SRR22245384
SA68	Brazil	1990	CP110192 ^B^
4295/02	Brazil	2002	SRR17885111
287/91	Brazil	2010	SRR21618241
SG9-1955	UK	1955	CM001153-CM001154 ^B^
SG9-1995	UK	1995	SRR1045136
ST572	UK	2009	SRR2121409
11CEB2315SAL	France	2011	ERR9714929
SARB21 ^A^	USA	1972	ERR424914
CDC 4801/72 ^A^	USA	1972	SRR1122702
ST 78 Tanzania	Tanzania	2017	SRR11005774
VTCCBAA614	India	2012	JSWQ00000000 ^B^

^A^ Strains isolated from humans. ^B^ GenBank accession number of assembled genomes.

**Table 3 genes-14-00823-t003:** Genomic statistics of Colombian and retrieved SG strains were analyzed in this study.

Strain	Genome Status	No. of Bases	No. of Contigs	N50	Depth	% GC	No. of CDS	MLST
Buga	Draft	4,703,728	43	290,630	58×	53.1	4509	78
GDX7	Draft	4,703,746	41	243,065	61×	51.5	4506	78
FTA303	Draft	4,769,563	140	163,753	77×	50.1	4589	78
GDX58	Draft	4,705,375	49	401,050	70×	51.1	4593	78
GDX2	Draft	4,704,264	34	400,863	70×	51.9	4505	78
GDX96	Draft	4,704,639	39	400,863	74×	51.2	4506	78
GDX-03965-17	Draft	4,702,114	56	181,715	60×	52.3	4506	78
GDX-03399-17	Draft	4,701,504	81	116,199	65×	51.7	4509	78
9R-Col	Draft	4,738,050	175	206,437	49×	51.9	4534	78
SA68	Complete	4,657,435	NA *	NA *	NA *	52.2	4412	78
4295/02	Draft	4,477,714	37	283,382	36×	52.3	4372	78
287/91	Complete	4,658,697	NA *	NA *	NA *	52.2	4510	331
SG9-1955	Complete	4,658,698	NA *	NA *	NA *	52.2	4392	78
SG9-1995	Draft	4,701,135	62	195,494	70×	50.1	4502	78
ST572	Draft	4,641,016	31	495,945	67×	51.7	4418	78
11CEB2315SAL	Draft	4,614,697	41	284,067	148×	51.7	4400	78
SARB21	Draft	4,560,140	47	235,893	32×	51.8	4336	78
CDC 4801/72	Draft	4,757,193	264	185,274	184×	50.9	4334	78
ST 78 Tanzania	Draft	4,737,077	35	403,192	113×	51.5	4549	78
VTCCBAA614	Draft	4,701,135	62	195,494	70×	50.1	4504	78

* NA: raw data were not available for assembly.

## Data Availability

The raw sequenced reads were deposited at the National Center for Biotechnology Information Sequence Read Archive database (SRA) under accession numbers SRR8521219 (Buga), SRR8521149 (GDX7), SRR8521220 (FTA303), SRR8521221 (GDX58), SRR22245387 (GDX2), SRR22245388 (GDX96), SRR22245385 (GDX-03965-17), SRR22245386 (GDX-03399-17), and SRR22245384 (9R-Col). The assembled genome sequences were deposited in GenBank under the BioProject number PRJNA483415 under accession numbers RHEL00000000 (Buga), QYTY00000000 (GDX7), QZND00000000 (FTA303), QYTX00000000 (GDX58), RHEK00000000 (GDX2), RHEJ00000000 (GDX96), QYUA00000000 (GDX-03965-17), QYTZ00000000 (GDX-03399-17), and RHEM00000000 (9R-Col).

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
