# Peer review of "Genomic Characterization and Genetic Profiles of Salmonella Gallinarum Strains Isolated from Layers with Fowl Typhoid in Colombia"

_genes, 2023, doi:10.3390/genes14040823_

Round 1

Author Response

Dear Editors and Reviewers:

We thank the reviewers for their thoughtful comments and constructive suggestions concerning our manuscript entitled “Genomic Characterization and Genetic Profiles of Salmonella Gallinarum Strains Isolated from Layers with Fowl Typhoid in Colombia” (ID: genes-2242370), which enabled us to resubmit a clearly improved manuscript. We highlighted the amendments in the revised manuscript, and responded, point by point to, the comments as listed below.

Reviewer #1:

Chacón et al., have characterized eight columbian Salmonella Gallinarum Strains which were Isolated from Layers with Fowl Typhoid. They have identified an inventory of antibiotic resistance genes, genomic islands, prophages and virulence factors in twenty SG strains. I have found that at some places clarity is missing. In addition, for few sections more information is also needed from authors.  I have some concerns regarding the study that are mentioned below.

R0. We greatly appreciate for the comments and suggestions on our manuscript.

Q1. Line 57-60 “Therefore, the systemic response it elicits, in contrast to an infection that primarily triggers an inflammatory response, facilitates invasion. [3,5]. In addition, its immobility, caused by the absence of flagella, renders it unable to activate the Toll-like receptor 5 (TLR5) response [4].” is not clear . Please explain it more in lucid manner

R1. We thank the reviewer for this observation. These lines were summarized and rephrased to better explain them as suggested.

Q2. Line 61 - (Define TF)

R2. We thank the reviewer for this observation. The proper abbreviation is FT which was defined on line 49. We apologize for the mistyping.

Q3. Vaid et al (Reference no. 23) has not focussed their study on decoding the differences between bv Gallinarum and bv Pullorum strains but they have compared the strains belonging to servoar Gallinarum. Put the reference in proper context.

R3. We thank the reviewer for this observation. We added the context of the reference as suggested.

Q4. 98 -100 line are not clear. Please mention that you have identified genetic features related to resistome, virulome and mobiliome of investigated SG genomes. 

R4. We thank the reviewer for this observation. We modified the text and included the genetic features identified as suggested.

Q5. Line 136 (NCBI SRA instead of SRA)

R5. We thank the reviewer for this observation. We modify the acronym as suggested.

Q6. Rewrite line 139-141 for better clarity.

R6. We thank the reviewer for this observation. We have rephrased the observed lines to better explain them.

Q7. Divide the lines 156-165 for clarity.

R7. We thank the reviewer for this observation. We divided the lines as suggested.

Q8. Were there any differences in the results obtained by the usage of CARD and RESfinder ? Mention in the results.

R8. We thank the reviewer for this observation. We chose CARD database because it included both chromosomal and acquired genes. On the other hand, we used the ResFinder v4.1 tool exclusively to look for chromosomal point mutations associated with AMR. We included these reasons into the 3.2 section of the results.

Q9. Clearly mention the status of genomes investigated (complete or draft) Please mention in Table 3/ manuscript.

R9. We thank the reviewer for this observation. We indicated the status of the genomes in the 2.3 section of the Materials and Method, and in the Table 3 ans section 3.2 of Results.

Q10. Subheading (Line 181) should also be related to phylogeny

R10. We thank the reviewer for this observation. We included a sentence in the 3.2 section of the manuscript to explain better the use of the genomes despite raw data was not available for three of them.

Q11. SPI-1, one of the most important pathogenicity islands in Salmonella is absent in strain 4295/02 along with SPI-13 and C63PI as per your results. I believe that you should also do manual BLASTN searches to reconfirm as well to increase the credibility of your results. In a similar manner you should also check the other SG strains in which presence of SPI-1 and SPI-7  is not being shown as per SPIfinder results to .

R11. We thank the reviewer for this observation. We have now performed manual searches in Blast to confirm the presence/absence of the SPIs. We have updated the results, figures, tables, and discussions concerning these changes. We regret the previous discrepancies.

Q12. More information should be added about identified SPI’s such as location, percent identity among others and should be placed in supplementary file.

R12. We thank the reviewer for this observation. We now included a supplementary table including the additional information about Salmonella Pathogenicity Islands identified by SPIFinder v2.0 (Table S2).

Q13. Line 272 - is there anything missing ?  

R13. We thank the reviewer for this observation. We have rewritten the sentences for better clarity.

Q14. Line 285-297 please provide the detailed information (location, score and length etc) of the identified prophages and put them in supplementary file.

R14. We thank the reviewer for this observation. We now included a supplementary table including the additional information about Prophages identified by Phaster search tool (Table S4).

Q15. Mention the differences that were uniquely found in Columbian SG strains and not in other SG strains as well as the commonalities in the last section of discussion.

R15. We thank the reviewer for this observation. We included a paragraph with the similarities and differences in the Colombian strains with respect to the other SG strains, and the potential implications of this findings.

Q16. Is there any major difference or similarity of your study in which you have specifically taken Columbian SG strains and previous studies.

R16. We thank the reviewer for this observation. We included two sentences to refer to previous SG studies.

Q17. Please mention some methods in the manuscript by which this study can help to better understand the mechanisms underly SG virulence?

R17. We thank the reviewer for this observation. We included a paragraph listing the methods used in this study help to better understand the mechanisms underly SG virulence

Reviewer 2 Report

Comments to the Author

In the study, the authors sequenced the genomes of eight Colombian strains and one vaccine strain of Salmonella Gallinarum (SG), the agent of fowl typhoid. They compared them with other strains from different regions and analyzed their resistance, virulence, and mobile genetic elements. They found common features such as efflux pumps, gyrB mutations, SPIs, and prophages that could be related to the pathogenicity and evolution of this serotype.

Overall, the manuscript provides useful data to enhance the understanding of the genomic profile of SG, especially for the ‘Colombian file isolates’. However, the number of SG strains analyzed was limited (more than 100 SG WGS data are available in NCBI) and the discussion part should focus more on clarifying the features of the Colombian field strains, as suggested by the manuscript title. Some specific points for potential improvement are listed below:

Line 30: Write “Salmonella” in italics (the same for other related places).

Line 56-66: This paragraph seems to be tangential to this study. Consider summarizing this content into the first paragraph.

Line 67-79: This paragraph is only to introduce the readers to the virulence-related genomic features of Salmonella. Try to revise it and add some clear sentences to explain that the importance of virulence-related factors in Salmonella is the motivation behind this study’s aim to examine their distribution among different SG isolates.

Line 80-88: This paragraph needs to be improved. You should clarify to the readers that the reason you included the 9R strain in the analysis is to understand the genomic variation between this vaccine strain and the other field SG strains, which may provide insights into SG virulence.

Figure 2: What does TH mean? Please add the relevant information.

Line 226: it is impossible to see the SNPs in Figure 2.

Line 230-232: This is a method rather than a result. What does 95% refer to? Identity or coverage? Do you mean the following categories: complete: coverage = 100%; incomplete (1): 100%>coverage >60%; incomplete (2): 60%>=coverage >20%?

Figure 4: The figure is hard to see clearly; a supplementary table would be helpful.

Line 330-332: Considering the aim of this study, the following discussion should provide a more detailed analysis of the ‘Colombian field strains’ and their distinctive features. Any potential implications?

Line 340-341: To my understanding, there are many Salmonella WGS-based studies (including the gene-character comparison).

Line 386-387: Why is this interesting? Does 4295/02 show any phenotypes related to SPI-1 deletion?

Line 389-404: These contents are only an introduction to SPIs; I cannot see any relevant discussion for them. These should be removed.

Line 408-409: Add the reference.

Line 462-470: The conclusion paragraph seems to be weak, and it would be better to include some additional points that briefly summarize the study. The current content is mainly composed of vague generalities that do not convey the main findings and contributions of the study.

Author Response

Dear Editors and Reviewers:

We thank the reviewers for their thoughtful comments and constructive suggestions concerning our manuscript entitled “Genomic Characterization and Genetic Profiles of Salmonella Gallinarum Strains Isolated from Layers with Fowl Typhoid in Colombia” (ID: genes-2242370), which enabled us to resubmit a clearly improved manuscript. We highlighted the amendments in the revised manuscript, and responded, point by point to, the comments as listed below.

Reviewer #2:

In the study, the authors sequenced the genomes of eight Colombian strains and one vaccine strain of Salmonella Gallinarum (SG), the agent of fowl typhoid. They compared them with other strains from different regions and analyzed their resistance, virulence, and mobile genetic elements. They found common features such as efflux pumps, gyrB mutations, SPIs, and prophages that could be related to the pathogenicity and evolution of this serotype.

Overall, the manuscript provides useful data to enhance the understanding of the genomic profile of SG, especially for the ‘Colombian file isolates’. However, the number of SG strains analyzed was limited (more than 100 SG WGS data are available in NCBI) and the discussion part should focus more on clarifying the features of the Colombian field strains, as suggested by the manuscript title. Some specific points for potential improvement are listed below:

R0. We greatly appreciate for the comments and suggestions on our manuscript.

Q1. Line 30: Write “Salmonella” in italics (the same for other related places).

R1. We thank the reviewer for this observation. We italicize the word throughout the manuscript as suggested.

Q2. Line 56-66: This paragraph seems to be tangential to this study. Consider summarizing this content into the first paragraph.

R2. We thank the reviewer for this observation. These lines were summarized and rephrased to better explain them as suggested.

Q3. Line 67-79: This paragraph is only to introduce the readers to the virulence-related genomic features of Salmonella. Try to revise it and add some clear sentences to explain that the importance of virulence-related factors in Salmonella is the motivation behind this study’s aim to examine their distribution among different SG isolates.

R3. We thank the reviewer for this observation. We modified this paragraph and added some sentences to explain better the importance of studies of virulence factors among SG isolates.

Q4. Line 80-88: This paragraph needs to be improved. You should clarify to the readers that the reason you included the 9R strain in the analysis is to understand the genomic variation between this vaccine strain and the other field SG strains, which may provide insights into SG virulence.

R4. We thank the reviewer for this observation. We modified this paragraph and added a couple of sentences to explain better the importance of genomic studies comparing 9R and field isolates.

Q5. Figure 2: What does TH mean? Please add the relevant information.

R5. We thank the reviewer for this observation. We replaced that abbreviation by the term “coverage”.

Q6. Line 226: it is impossible to see the SNPs in Figure 2.

R6. We thank the reviewer for this observation. We modified this figure to include the mutations in gyrase genes associated with antibiotic resistance.

Q7. Line 230-232: This is a method rather than a result. What does 95% refer to? Identity or coverage? Do you mean the following categories: complete: coverage = 100%; incomplete (1): 100%>coverage >60%; incomplete (2): 60%>=coverage >20%?

R7. We thank the reviewer for this observation. Those lines were transferred to the respective material an methos sections. We also rephrased them to explain better.

Q8. Figure 4: The figure is hard to see clearly; a supplementary table would be helpful.

R8. We thank the reviewer for this observation. We now included a supplementary table including all the virulence factors detected in the studied strains (Table S3).

Q9. Line 330-332: Considering the aim of this study, the following discussion should provide a more detailed analysis of the ‘Colombian field strains’ and their distinctive features. Any potential implications?

R9. We thank the reviewer for this observation. We included a paragraph with the similarities and differences in the Colombian strains with respect to the other SG strains, and the potential implications of this findings.

Q10. Line 340-341: To my understanding, there are many Salmonella WGS-based studies (including the gene-character comparison).

R10. We thank the reviewer for this observation. We rephrased these lines as suggested.

Q11. Line 386-387: Why is this interesting? Does 4295/02 show any phenotypes related to SPI-1 deletion?

R11. We thank the reviewer for this observation. We rephrased these sentences to expose better the potentials implications over the virulence of that strain.

Q12. Line 389-404: These contents are only an introduction to SPIs; I cannot see any relevant discussion for them. These should be removed.

R12. We thank the reviewer for this observation. We removed those lines as suggested.

Q13. Line 408-409: Add the reference.

R13. We thank the reviewer for this observation. We added the reference to this sentence.

Q14. Line 462-470: The conclusion paragraph seems to be weak, and it would be better to include some additional points that briefly summarize the study. The current content is mainly composed of vague generalities that do not convey the main findings and contributions of the study.

R14. We thank the reviewer for this observation. We modified the conclusion paragraph to include points summarizing the study.

Round 2

Reviewer 2 Report

I am fine with the response and revision.

Author Response

We thank the reviewers for their thoughtful comments and constructive suggestions concerning our manuscript entitled “Genomic Characterization and Genetic Profiles of Salmonella Gallinarum Strains Isolated from Layers with Fowl Typhoid in Colombia” (ID: genes-2242370), which enabled us to resubmit a clearly improved manuscript.